# Research on Recycling of Phosphorus Tailings Powder in Open-Graded Friction Course Asphalt Concrete

**DOI:** 10.3390/ma16052000

**Published:** 2023-02-28

**Authors:** Yue Xiao, Xinjiang Ju, Chao Li, Tianlei Wang, Rui Wu

**Affiliations:** 1State Key Laboratory of Silicate Materials for Architectures, Wuhan University of Technology, Wuhan 430070, China; 2Green Building Materials and Manufacturing Engineering Research Center of the Ministry of Education, Wuhan University of Technology, Wuhan 430070, China; 3Section of Pavement Engineering, Faculty of Civil Engineering & Geosciences, Delft University of Technology, Stevinweg 1, 2628 CN Delft, The Netherlands

**Keywords:** phosphorus tailings, asphalt mixture, solid waste recycling, high-temperature rheological properties, moisture damage resistance

## Abstract

The reuse in high-value materials is one of the important resource utilization approaches of phosphorus tailings. At present, a mature technical system has been formed on the reuse of phosphorus slag in building materials, and silicon fertilizers in the extraction of yellow phosphorus. But there is a lack of research on the high-value reuse of phosphorus tailings. In order to make safe and effective utilization of phosphorus tailing resources, this research concentrated on how to solve easy agglomeration and difficult dispersion of phosphorus tailing micro-powder, when it was recycled in road asphalt. In the experimental procedure, phosphorus tailing micro-powder is treated in two methods. One method is to directly add it with different contents in asphalt to form a mortar. Dynamic shear tests were used to explore the effect of phosphorus tailing micro-powder on the high-temperature rheological properties of asphalt influence mechanism of material service behavior. The other method is to replace the mineral powder in asphalt mixture. The effect of phosphate tailing micro-powder on the water damage resistance in open-graded friction course (OGFC) asphalt mixtures was illustrated, based on the Marshall stability test and the freeze–thaw split test. The research results show that the performance indicators of the modified phosphorus tailing micro-powder meet the requirements for mineral powder in road engineering. Compared with standard OGFC asphalt mixtures, the residual stability of immersion and freeze–thaw splitting strength were improved when replace the mineral powder. The residual stability of immersion increased from 84.70% to 88.31%, and freeze–thaw splitting strength increased from 79.07% to 82.61%. The results indicate that phosphate tailing micro-powder has a certain positive effect on the water damage resistance. These performance improvements can be attributed to the larger specific surface area for phosphate tailing micro-powder than ordinary mineral powder, which can effectively adsorb asphalt and form structural asphalt. The research results are expected to support the large-scale reuse of phosphorus tailing powder in road engineering.

## 1. Introduction

Phosphate rock is a typical kind of sedimentary rock that contains high amounts of phosphate minerals, widely used in fertilizer manufacture and other industries [1]. China is reviewed as the second largest store of phosphate rocks in the world according to the data from the International Fertilizer Industry Association. In recent years, the overuse of phosphate rock has led to an increasing amount of green gas and solid wastes, which has become a big threat to the ecological environment [2].

Phosphorus tailings are a typical by-product [3] in phosphate mining and flotation processes. When 1 ton of phosphorus concentrate is produced, it will generate 0.4 tons [4,5] of phosphorus tailings. There are nearly 9.5 million tons phosphorus tailings accumulated every year in China. However, only a limited amount of them can be reused, with an efficiency of about 7% [6]. The stacking of phosphate tailings will bring heavy metals into the soil or water source and threaten human physical health. Research from K. Gnandi [7,8] shows that phosphate tailings contain heavy metal elements such as Cr, Cd, Cu, V, Ni, U, and Zn which could increase the risk of illness. Meanwhile, heavy metals will accrue in crustaceans and fish through streams. The effect will be more obvious at a closer distance from the phosphorus tailings. Therefore, the question of how to reuse phosphorus tailings in a safer and more effective approach is significant and urgent.

The main chemical components in phosphate tailings are considered [9,10,11] to be silicon dioxide (SiO_2_), magnesium oxide (MgO), iron oxide (Fe_2_O_3_), phosphorus pentoxide (P_2_O_5_), calcium oxide (CaO), and others. Therefore, phosphate tailings are generally recycled [12] in construction materials such as cement and ceramics. Adding phosphate tailings in the production of sintered brick can enhance its structure performance, and phosphate tailings can replace fine aggregates in cement concrete [13]. Moreover, the phosphate tailing content will affect the properties of the setting time and hydration heat of cement concrete. The increasing of the phosphate tailing content will extend the setting time and decrease the hydration heat [14]. In addition, phosphate tailings are a suitable resource for fertilizer, since the silicon, phosphorous, and calcium are beneficial in agriculture.

In pavement engineering, industrial solid wastes are processed into a microfine powder and then directly added to asphalt as a modifier or as a replacement of mineral powder [15]. In general, the addition of inorganic powder as an additive or modifier will bring an excellent effect in improving asphalt performance such as fly ash, silica fumes, slag, and so on [16]. Sharma [17] has improved the growth of strength when waste marble powder is used in place of sand and cement in concrete. However, limited attention has been focused on the application of phosphate tailings in asphalt concrete. Previous researchers mainly were concerned with phosphate slag, another type of solid waste in phosphate. Qian [18,19] found that phosphate slag powder is alkaline and hydrophobic at high temperatures, which brought an improvement in the anti-aging and rutting resistance properties for virgin asphalt, as well as high-temperature stability and water resistance performance for asphalt mixture. Sheng [20] investigated the influence of phosphate slag on the rheological properties and mechanical properties. It appeared reasonable to replace the conditional filler with phosphorus slag powder. Wang [21] used phosphate slag as the modifier in base asphalt. Several experiments such as Marshall tests, rutting tests, and water immersion Marshall tests are conducted to investigate its influence. The results showed that phosphate-slag-modified asphalt will reach its best high-temperature stability and water stability at a 9% dosage amount. The SEM and FTIR results are consistent with the mechanic performance where the addition of phosphate slag is beneficial for road performance. Therefore, the research methods for phosphate slag can be adopted for phosphate tailings on account of their similar composition.

At present, it will be a novel method in phosphorus tailing utilization. Although there are only a few works of research on the effect of phosphate tailings on the moisture susceptibility of dense graded asphalt mixture [22], there is a lack of attention on open-graded friction course (OGFC) asphalt mixtures which are more susceptible to moisture damage [23,24,25]. The OGFC asphalt mixture has more void content. After the rain, the OGFC mixture will store more water in its structure and is more fragile with moisture. Therefore, research on OGFC has a more significant application. The main objective of this research is to explore the influence of phosphate tailing microfine powder on moisture damage resistance in the OGFC asphalt mixture.

## 2. Materials and Methods

Phosphate tailing micro-powder was used as the modifier for the asphalt binder and replacement for traditional limestone mineral powder after proper treatment. To achieve this goal, a series of experiments was made which involved characterization of phosphate tailing powder, asphalt mortar, and OFGC mixture. For phosphate tailing powder, experiments are concentrated on its chemical composition, surface structure, and pore properties. For asphalt mortar, physical properties (penetration, softening point, and extensibility) and dynamic shear rheometer (DSR) test are designed for its road performances. For OGFC mixture, Marshall water immersion test and freeze–thaw split test are performed to measure its water damage resistance. The research flow is shown in Figure 1.

### 2.1. Phosphate Tailing Pretreatment

Owing to the original hydrophobic and oleophobic surface of phosphate tailings and different polarity of inorganic powder and organic asphalt and hydrophobic and oleophobic surface, this will inform the inhomogeneous mixture when phosphate tailing powder is mixed with the asphalt binder. The mixture is not stable and under the risk of segregation. Thus, some necessary procedures will be made to improve compatibility of asphalt and phosphate tailings. Wang [26] had compared five different modifiers based on wetting test, oil absorption value, and dispersion test, and concluded that phosphate type mono-alkoxy titanate is the best choice for phosphate tailing modification. In this research, phosphate type mono-alkoxy titanate was used for phosphate tailing modification and the final powder appearance is shown in Figure 2a. The color of the final powder was grey and the powder did not clump after drying. There are several requirements for filler in asphalt mixture from Technical Specification for Construction of Highway Asphalt Pavements (JTG F40-2004). After pretreatment, the modified phosphate tailing powders satisfied all filler requirements, which is shown in Table 1. The apparent relative density was measured by a Lee’s specific gravity bottle which is shown in Figure 2b. The pH test is based on standard test methods of mineral powder pH for asphalt pavements. The 10 g dried phosphate tailing powder was added to 50 mL purified water, followed by the steps of shaking, stirring, and keeping it in 30 min. The pH of final solution is weak alkaline in 9.3, resulting in positive compatibility between asphalt and phosphate tailings.

### 2.2. Modifier in Asphalt Mortar

Although phosphorus tailing powder was usually added as a substitute for mineral powder [27,28], attention is also needed on its influence when phosphorus tailing powder acted as an extra modifier. Therefore, this part of the experiment was designed only for phosphate tailing powder influence on the asphalt binder instead of hot mix asphalt. Phosphorus tailing powder was added into high-viscosity asphalt with mass percentages of 4%, 7%, 10%, 12%, and 15%. The asphalt binder is the same type in the following OGFC asphalt mixture.

The detailed preparation processes are listed as follows: At first, modified phosphorus tailing micro-powder and the asphalt binder were heated in oven at 170 °C for 4 h. Then, phosphate tailing powder was added to the preheated asphalt binder in different proportions with constant stirring. Next, the initial mix of asphalt and phosphorus tailing powder was cycle-sheared by a high-speed shearing instrument for 30 min in 170 °C, and sheared at slow speed for 5 min to discharge all the air bubbles. Finally, the asphalt was put in oven at 150 °C for 1 h to obtain the final product.

### 2.3. Ingredient Replacement in OGFC Asphalt Mixture

Porous asphalt mixture is widely used in urban pavements with great permeability. It can enhance safety for driving and increase the anti-slip ability of the road. At the same time, porous asphalt mixture is useful in reducing the occurrence of glare at night by leaking water on the pavement surface. However, large amount of internal semi-connected void structure prevents drainage [29], raising the risk of moisture damage. Therefore, OGFC-16 asphalt mixture [30,31] is chosen with which to discuss the influence of phosphate tailing powder on water damage resistance, because OGFC will face a bigger challenge on rainy days. This experiment part was designed for asphalt mixture. Phosphate tailing powder is used as the replacement for mineral powder.

Asphalt mixture gradation was shown in Table 2. The appropriate oil/stone ratio was determined as 4% and void ratio of mixture was measured as 22.8%. Coarse and fine aggregates used in the mix were limestone. In addition, limestone powder and phosphorus tailing powder were, respectively, used for fillers. There are two OGFC-16 asphalt samples in total. One of them 100% replaces limestone powder by phosphorus tailing powder, and one of them uses ordinary limestone powder as filler for comparison.

## 3. Results and Discussion

### 3.1. Performances of Pretreated Phosphate Tailings

#### 3.1.1. Chemical Composition

Chemical analysis is conducted via X-ray diffraction (XRD) using an Advance D8 (Bruker, Bremen, Germany). The resulting XRD pattern is shown in Figure 3. It shows that there are two main compositions in phosphate tailing micro-powder, which are recognized as Ca_5_(PO_4_)_3_F and CaMg(CO_3_)_2_, respectively.

#### 3.1.2. Surface Structure

The surface structure is observed from scanning electron microscopy (SEM) using a JSM-IT300 (Japan Electron Optics Laboratory, Tokyo, Japan). The morphology of the phosphorus tailing micro-powder is shown in Figure 4. It is illustrated in Figure 4 that the diameter of most of the powder is distributed less than 100 μm with a rough surface and regular shape. The rough surface is believed to enhance the adhesion of the phosphorus tailings with asphalt, resulting in a positive contribution on the adhesion strength between the powder and aggregates/binder.

#### 3.1.3. Size Distribution

The size distribution is measured by a Mastersizer 2000 laser particle size analyzer (Malvern, Worcestershire, United Kingdom), and the results are shown in Table 3 and Figure 5. The main diameter sizes of the phosphorus tailings are between 10–100 μm, which is consistent with the SEM image discussed in Section 3.1.2. The average particle size is 24 μm. The surface area of the micro-powder is 1.4 m^2^/g, larger than normal mineral powder used in the asphalt mixture. Therefore, the phosphorus tailing micro-powder could have a bigger area to connect with the asphalt binder to form a more stable structure. From the view of the surface area, the addition of phosphorus tailings is beneficial to temperature stability.

### 3.2. Characterization of Phosphate-Tailing-Modified Asphalt Mortar

#### 3.2.1. Basic Physical Properties

The penetration, softening point, and ductility of the asphalt binder are considered as the most important three properties in classification [32]. The results of the five sample groups are compared with the initial level (0%) and standard level (specification requirement) in Figure 6. Penetration and ductility are more sensitive to the phosphate tailing powder content, in contrast to the slight fluctuations in the softening point. Penetration and extensibility gradually decrease with the increase of the phosphate tailing powder percentage. Extensibility has a further fall than penetration, which are about 13% and 29%, respectively.

#### 3.2.2. Rheological Performance

The phosphate tailing powder will bring variations to the mechanic performance of the asphalt binder affected by a different mass percentage. Focusing on the impact of the addition of phosphate tailings on the rheological properties, the DSR test is used for a high-temperature rheological properties evaluation. The DSR test contains two models, the temperature sweep test and frequency sweep test, targeting the complex shear modulus (G*) and rutting factor (G*/sinδ). The composite shear modulus G* represents the asphalt’s ability to resist flow deformation, and the rutting factor G*/sinδ is commonly used to evaluate the rutting resistance of asphalt [3]. The specimens are the basic asphalt binder, and five powder–asphalt mortars with a 4%, 7%, 10%, 12%, and 15% mass percentage. The picture of the DSR instrument and six mixture specimens is shown in Figure 7.

In the temperature sweep test, the load type is settled in the strain-controlled mode with the strain value γ = 12%, and the temperature range is between 46 to 82 °C with the loading frequency ω = 10 rad/s. The variation of the complex shear modulus (G*) is shown in Figure 6, as well as the rutting factor (G*/sinδ) in Figure 7.

As shown in Figure 8A, for both the basic binder and powder–asphalt mortars, there exists an inverse relationship between the G* and temperature. With the increase of the test temperature, the G* of all specimens continuously declines. The decline speed of G* is fast at the beginning and slows down as the temperature rises. As for the decrease ratio, the higher phosphate tailing mass percentage specimen has a more obvious drop. In hot summer months, the temperature of the asphalt pavement surface can exceed 60 °C. However, starting from 64 °C to 70 °C, 76 °C, and 82 °C, the change in G* for all specimens is relatively small. Therefore, the resistance to flow deformation decreases with the temperature decrease for the basic binder and powder–asphalt mortars. In Figure 8B, the influence of the mass content on G* differs at different temperatures. At relatively lower temperatures (46 °C and 52 °C), G* noticeably increases with the increase of the powder mass content. The G* of the 15% phosphate tailing powder is nearly 1.4 times that of basic asphalt. However, beyond 64 °C, there is no visual change in G*. The results are thought to show that mass percentages have an obvious impact on the resistance to flow deformation at a lower temperature (under 64 °C) and has a limited impact at a higher temperature. In general, a larger mass percentage will bring better performance in resisting flow deformation. It does not show the peak at our test range (0–15%). For economy, more reuse of phosphate tailings will save more of the costs in pollution treatment. Therefore, there is potential for phosphate tailing reusage in asphalt mixtures.

In terms of G*/sinδ, Figure 9A illustrates a similar trend between G*/sinδ and temperature. With the increase of the test temperature, the G*/sinδ of all specimens continuously declines, which clarifies that the temperature increase will lead to a high-temperature stability reduction for all specimens. As with G*, changes in the G*/sinδ for all specimens are relatively small above 64 °C. It indicates that asphalt mortars almost lose their high-temperature stability because there is nearly no difference in the ability of flow deformation resistance in a higher temperature. In Figure 9B, G*/sinδ quickly develops with the mass content increasing under 64 °C, showing a great improvement in high-temperature stability. However, this progress is difficult to observe at high temperatures (over 64 °C). The results are thought to show that the addition of phosphate tailing powder will enhance the rutting resistance of asphalt mortars, but the effect varies at different temperatures. In general, a larger mass percentage will bring better performance in rutting resistance, which is the same as the result in G*.

In the frequency sweep test, the load type is settled in the strain-controlled mode with the strain value γ = 12%, and the frequency range is between 0.1 to 100 rad/s with the loading temperature 64 °C. The variation of G* is shown in Figure 10, as well as G*/sinδ. The variation of G* and G*/sinδ show a similar tendency.

It is indicated from Figure 8A that, under low-frequency loading, the curves of G* in all specimens stay on the same level with limited growth. In the meantime, for individual specimens, the value of G* remains stable with frequency growth under low-frequency loading. Therefore, it is concluded that the addition of phosphate tailings did not improve resistance to deformation under low-frequency loading conditions, which leads to an increased risk in heavy, low-speed traffic load. Under high-frequency loading (beyond 10 rad/s), G* increases rapidly in all specimens, and the specimen with a higher phosphate tailing content displays a faster increase rate. A small content of phosphate tailing powder is effective in strengthening G* under high-frequency loading. Therefore, this means that addition of phosphate tailings improves the performance of deformation resistance under high frequency. However, the deformation resistance of the asphalt mortars will remain stable with no significant increase in phosphate tailing contents of more than 10%. For G*/sinδ in Figure 8B, the conclusion is the same as G*. The addition of phosphate tailings has no effect on G*/sinδ under low-frequency loading, while it significantly enhances G*/sinδ under high-frequency loading.

In general, the addition of phosphate tailings results in an improvement in rutting resistance and resisting flow deformation under two test conditions. However, increasing the mass content did not significantly enhance the improvement value.

### 3.3. Moisture Damage Resistance in OPGC-16

As written in Chapter 2.3, ‘Ingredients Replacement in Hot Mixed Asphalt’, the asphalt mixture with phosphate tailings as the mineral powder was used as the experiment group, while the asphalt mixture with ordinary limestone as the mineral powder was used as the basic group. The stability in the water immersion test and splitting strength after freeze–thaw cycles were the two main indicators in this research. The Marshall water immersion test and freeze–thaw split test were conducted, based on specification JTG E20-2011. The absolute results from the Marshall test and freeze–thaw split test are shown in Table 4. The residual stability and residual strength ratio are shown in Figure 11.

For the residual stability in the Marshall water immersion test, the basic group and experiment group all meet the requirement of 80% in the specification. The residual stability in the experiment group was increased from 84.70% to 88.31%, compared with the basic group, indicating that the addition of the phosphate tailing powder is beneficial for the water stability performance. For the residual strength ratio in the freeze–thaw split test, the basic group and experiment group all meet the requirement of 75% in the specification. The residual strength ratio in thet experiment group was increased from 79.09% to 82.61%, compared with the basic group, which is consistent with the results in the water immersion test.

## 4. Conclusions

The possibility of reusing phosphorus tailing powder in open-graded friction course asphalt concrete was tested and discussed in this research, so that the effect mechanism of modified phosphate tailing micro-powder on the basic physical properties and water damage resistance of asphalt mortar and OGFC mixture were reported. According to the reported data, the main conclusions can be drawn as listed:

(1) The phosphate tailing micro-powder is weakly alkaline. After a surface modification, its particle surface is rough and regular. Its main particle size ranges between 10–100 μm, with 24 μm as the average. Its surface area is 1.4 m^2^/g, more than common mineral powder, which is beneficial for the adsorption of the asphalt binder.

(2) For its influence in asphalt mortar, modified phosphate tailing micro-powder is homogeneously dispersed in asphalt, resulting in a positive contribution to high-temperature stability and rutting resistance. The G* and G*/sinδ of the 15% phosphate tailing powder content are both 1.4 times higher than those of the basic sample without phosphate tailing powder.

(3) For influence in OGFC asphalt mixture, modified phosphate tailing micro-powder can improve the resistance to water damage. In OGFC mixture, residual stability has risen from 84.70% to 88.31%, and residual strength ratio has risen from 79.07% to 82.61%. Based on the physical properties of phosphate tailing based mortar and moisture damage resistance of phosphate tailing based mixture, it can be concluded that modified phosphate tailing can be successfully reused in OGFC asphalt concrete.

The results show that phosphate tailing powders can surely improve the performance of asphalt and OFGC mixture, and strengthen its resistance to water damage. However, the current research lacks the explanation for this mechanism. This will restrict the design of further research plans. Therefore, future studies should expand the research scope. There are several suggestions, such as a trial for the exposure in real rainy weather, and more types of asphalt binder need to be included. Numerical model schemes should also be taken into account for a reliable illustration of the mechanisms.

## Figures and Tables

**Figure 1 materials-16-02000-f001:**
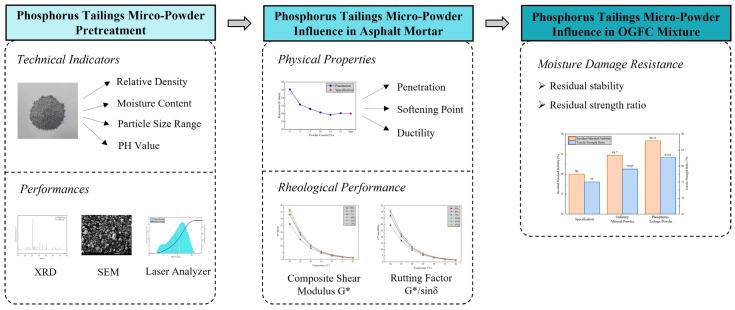
The research diagram of this study.

**Figure 2 materials-16-02000-f002:**
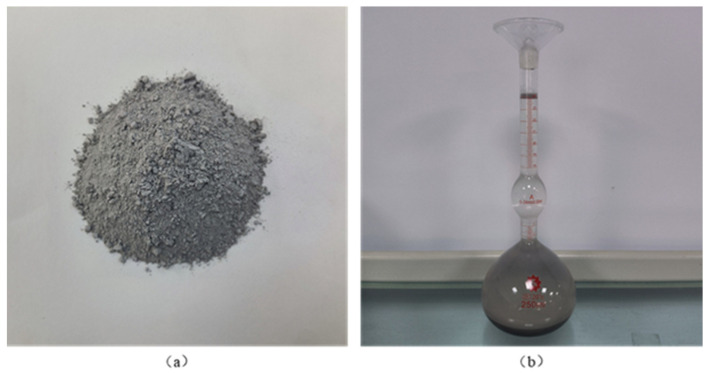
Pretreated phosphorus tailing micro-powder and Lee’s pycnometer. (**a**) Pretreated phosphorus tailing micro-powder; (**b**) Lee’s pycnometer.

**Figure 3 materials-16-02000-f003:**
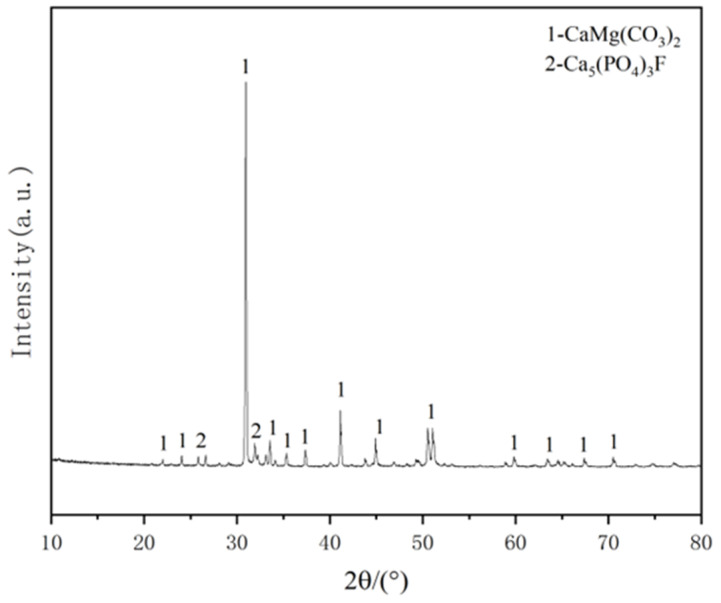
XRD pattern of phosphorus tailing micro-powder.

**Figure 4 materials-16-02000-f004:**
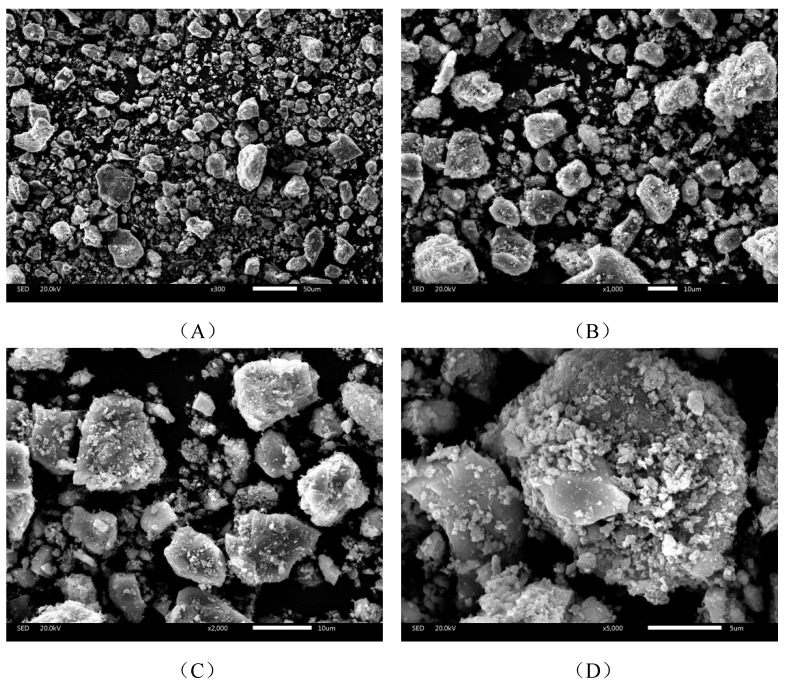
SEM images of phosphorus tailing micro-powder under different magnification: (**A**) 300 times, (**B**) 1000 times, (**C**) 2000 times, and (**D**) 5000 times.

**Figure 5 materials-16-02000-f005:**
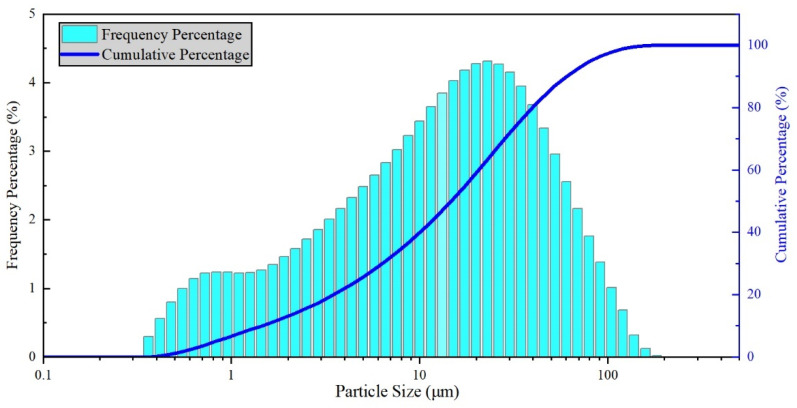
Particle size distribution curve of phosphorus tailing powder.

**Figure 6 materials-16-02000-f006:**
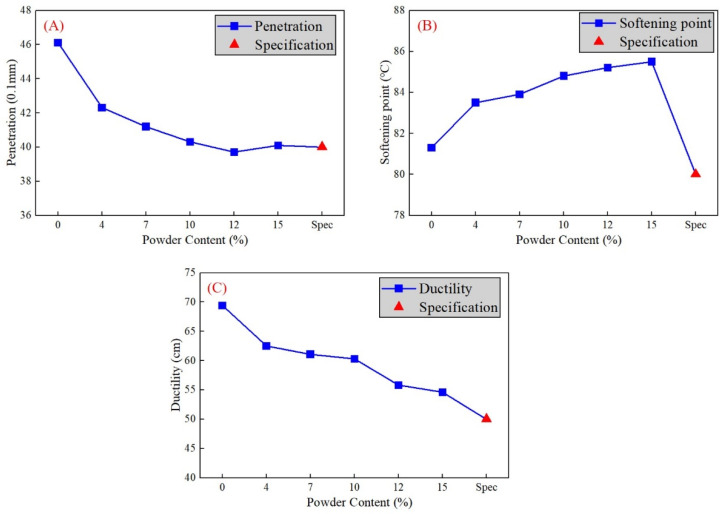
Variation trend of (**A**) penetration, (**B**) ductility, and (**C**) softening point with the increase of phosphorus tailing fine powder content.

**Figure 7 materials-16-02000-f007:**
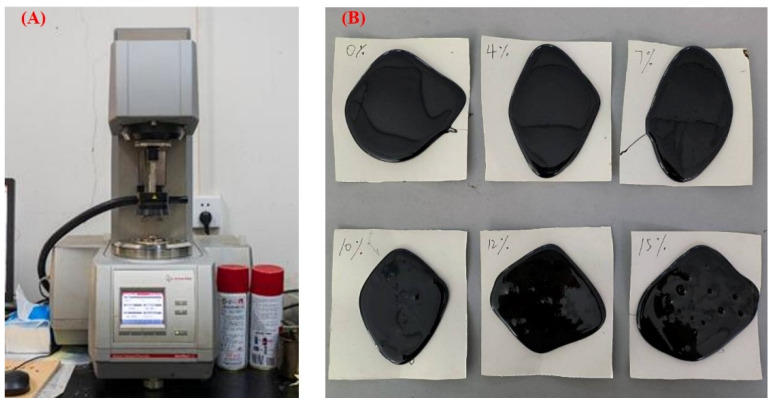
(**A**) DSR instrument; (**B**) modified asphalt specimens with different dosage.

**Figure 8 materials-16-02000-f008:**
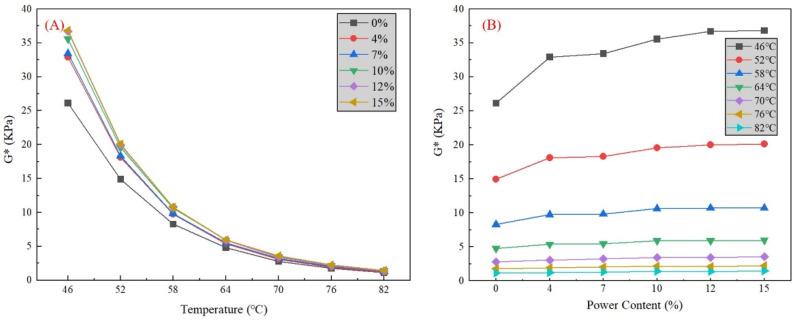
Variation of composite shear modulus G* with different (**A**) test temperature, and (**B**) mineral powder content.

**Figure 9 materials-16-02000-f009:**
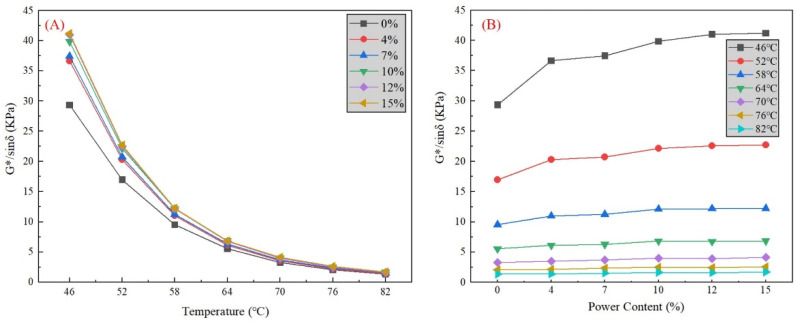
Variation of rutting factor G*/sinδ with different (**A**) test temperature, and (**B**) mineral powder content.

**Figure 10 materials-16-02000-f010:**
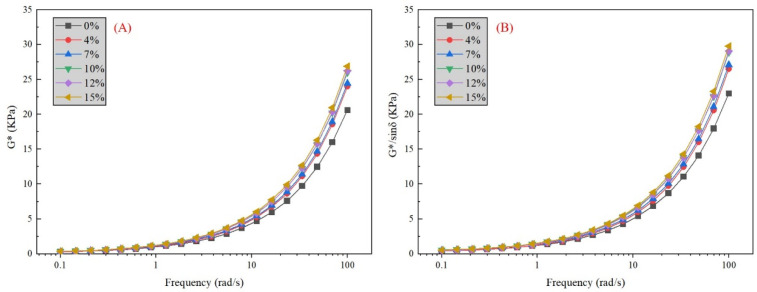
Different parameters varying with frequency: (**A**) composite shear modulus G*; (**B**) rutting factor G*/sinδ.

**Figure 11 materials-16-02000-f011:**
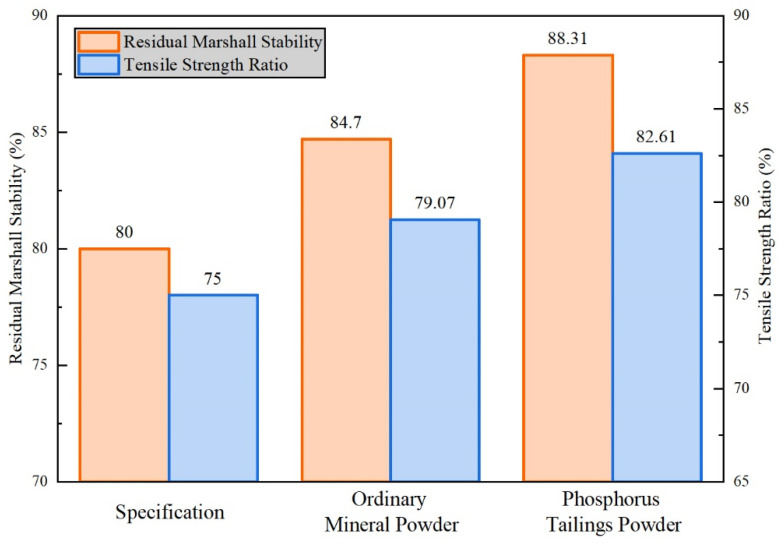
Comparison of moisture damage resistance of two asphalt mixtures.

**Table 1 materials-16-02000-t001:** Main technical indicators of phosphorus tailing powder.

Test Items		Test Result	Specification
Apparent relative density (g/cm^3^)		2.93	≥2.5
Moisture content (%)		0.3	≤1
Particle size range	<0.6 mm	100	100
	<0.15 mm	99.5	90–100
	<0.075 mm	92.5	75–100
Appearance (caking or not, after drying)		Not	Not
PH value		9.3	-

**Table 2 materials-16-02000-t002:** OGFC-16 asphalt mixture mineral gradation.

Aperture size (mm)	19	16	13.2	9.5	4.75	2.36	1.18	0.6	0.3	0.15	0.075
Passing percentage (%)	100	92	74	50	16	12	8	6	5	4	3

**Table 3 materials-16-02000-t003:** Particle size analysis results of phosphorus tailing micro-powder.

Volume Average Particle Size (μm)	Surface Area Average Particle Size (μm)	Specific Surface Area (m^2^/g)	Medium Particle Size (μm)
24.097	4.276	1.4	14.705

**Table 4 materials-16-02000-t004:** Results from Marshall test and freeze–thaw split test.

Mixture Type	Marshall Stability (kN)	Stability after 48 h Soaking (kN)	Residue Marshall Stability (%)	Unfrozen Splitting Strength (MPa)	Frozen Splitting Strength (MPa)	Tensile Strength Ratio (%)
Asphalt mixture with mineral powder	9.87	8.36	84.70	0.86	0.68	79.07
Asphalt mixture with phosphate tailings	9.92	8.76	88.31	0.92	0.76	82.61

## Data Availability

Not applicable.

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
