# Peer review of "Research on Recycling of Phosphorus Tailings Powder in Open-Graded Friction Course Asphalt Concrete"

_materials, 2023, doi:10.3390/ma16052000_

Round 1

Reviewer 1 Report

Comments:

1.     There needs to be a series of figures corrected. Figure 1 is an example of this repetition.

2.     What standard was applied to table 1?

3.     Why phosphorus tailings chosen rather than marble powder, glass fibre, steel, or polypropylene given that they greatly enhance the mechanical qualities of asphalt materials?

4.     Provide more information on the performance of other wastes in comparison to phosphorus tailings and cite the following relevant references:

·       DOI: 10.3390/pr10122745

·       DOI: 10.3390/ma15175811

5. Alternative fillers are often used in asphalt. The significant novelties of the research   should therefore be listed towards the end of the introduction. What is being looked at again to make the content worthwhile for publication?

6. It is advised that the conclusion section be expanded to include the study's numerical results, the suggested approach's limitations, and potential future research areas.

7. Information on the granulometry of the study materials, particularly alternate fillers, must be included in detail. Include this information in the revised manuscript, if possible.

Reviewer 2 Report

Enter a better way to cite publications in the text (Arabic numerals).

If possible, list more authors who have done similar studies on phosphate waste.

For the Marshall study also take absolute results: force, stability, deformation. What is this research for anyway? The ITSR Indirect tensile strength test is performed to assess water and frost resistance.

The set at the end compares the results for the DSR study.

Show photos of apparatus, test diagrams and samples.

Author Response

The reply is in attached Word file.

Reviewer 3 Report

The following comments should be considered while revising the manuscript.

·         Referencing style should follow according to the journal requirement.

·         Page 3, line 96 it is mentioned Wang had compared …. What does this mean? Is it from a reference?

·         The modification process used for phosphorus tailings micro-powder seems quite tedious. How realistic is this process for real application of this materials? Please discuss briefly.

·         Overall the percentages of improvement in different tests are not much significant when the performance of phosphate tailings powder was compared with ordinary mineral powder. So, some other analysis would be required before coming to any solid conclusion. What about the cost of these two asphalt mixtures? The authors should perform some cost analysis.

·         Conclusions section should improve further. There is nothing about basic physical properties and recommendations for future studies.

Author Response

The reply is in attached Word file.
